# Bose-Einstein condensation of non-ground-state caesium atoms

Milena Horvath[1], Sudipta Dhar[1], Arpita Das [2], Matthew D. Frye [3], Yanliang Guo [1], Jeremy M. Hutson [3], Manuele Landini [1] & Hanns-Christoph Nägerl [1] ✉

Bose-Einstein condensates of ultracold atoms serve as low-entropy sources for a multitude of quantum-science applications, ranging from quantum simulation and quantum many-body physics to proof-of-principle experiments in quantum metrology and quantum computing. For stability reasons, in the majority of cases the energetically lowest-lying atomic spin state is used. Here, we report the Bose-Einstein condensation of caesium atoms in the Zeeman-excited $m_f = 2$ state, realizing a non-ground-state Bose-Einstein condensate with tunable interactions and tunable loss. We identify two regions of magnetic field in which the two-body relaxation rate is low enough that condensation is possible. We characterize the phase transition and quantify the loss processes, finding unusually high three-body losses in one of the two regions. Our results open up new possibilities for the mixing of quantum-degenerate gases, for polaron and impurity physics, and in particular for the study of impurity transport in strongly correlated one-dimensional quantum wires.

Ultracold atomic gases have proven to be a fruitful testbed for few- and many-body quantum physics, in part due to their high degree of controllability[1]. A very powerful tool for the ultracold-atom platform is the ability to tune the interactions between the atoms via Feshbach resonances[2]. One atomic element that has been very successful in this regard is Cs[3,4]. The hyperfine ground state of Cs is enriched by an abundance of broad and narrow Feshbach resonances, and interaction tuning has been instrumental to a diverse series of seminal results on a wide range of topics. These include Bose-Einstein condensation (BEC)[3], Efimov physics[5,6], ultracold molecules[7–9], strongly correlated one-dimensional (1D) physics[10–12], long-range tunneling dynamics[13] and density-induced tunneling[14], scale invariance[15], matter-wave jets[16], and, recently, cooling by dimensional reduction[17] and the 1D-2D crossover[18]. These results have all been obtained by making use of one particular hyperfine Zeeman sublevel of Cs, the state $(f = 3, m_f = 3)$, which is the energetically lowest-lying Zeeman state.

BEC of atoms in excited states offers additional possibilities. Such condensates have been produced with excited Zeeman states, meta-stable electronic states, and atoms in higher bands of optical lattices. They have been used to create spinor quantum gases[19], to observe quantum droplet states[20] and to study unconventional superfluidity in excited lattice orbitals[21]. Early attempts to condense Cs in excited Zeeman sublevels were hindered by uncontrolled losses[22–26]. Later experiments using the sublevel (3, 3) benefited from the absence of inelastic two-body processes, but care was needed to avoid detrimental three-body collisions[3,5,27].

Here we report the achievement of a tunable Cs BEC in a state other than the absolute ground state, namely in the Zeeman-excited state $(f = 3, m_f = 2)$. We identify one particular window around a magnetic field of $B \approx 160$ G in which two- and three-body processes are sufficiently suppressed that pure condensates can reliably be produced with $3 \times 10^4$ atoms. In a second window around $B \approx 40$ G, partial condensation is possible. We find surprisingly high three-body losses

[1]Institut für Experimentalphysik und Zentrum für Quantenphysik, Universität Innsbruck, Technikerstraße 25, Innsbruck, Austria. [2]Joint Quantum Centre (JQC) Durham-Newcastle, Department of Physics, Durham University, Durham DH1~3LE, United Kingdom. [3]Joint Quantum Centre (JQC) Durham-Newcastle, Department of Chemistry, Durham University, Durham, United Kingdom. ✉e-mail: christoph.naegerl@uibk.ac.at

in this window, most likely due to the opening up of a new decay channel. Our work is guided by state-of-the-art coupled-channel calculations to determine the two-body scattering properties.

The attainment of BEC requires that the ratio of good to bad collisions is sufficiently high while effective one-body processes such as background-gas collisions and inelastic light scattering are negligible. Elastic collisions are needed to drive the evaporation and thermalization process, while two-body inelastic collisions and three-body recombination reduce the cooling efficiency, possibly to the point that BEC cannot be reached. With peak number densities in the range between $1 \times 10^{11}$ and $1 \times 10^{13}$ atoms/cm$^3$ during the cooling process, this translates into concrete values for the s-wave scattering length and into acceptable upper bounds for the two- and three-body loss-rate coefficients.

We have carried out coupled-channel calculations of the two-body scattering properties as a function of the magnetic field, as described in Ref. 28. The calculations use the interaction potential of Ref. 29 and a basis set including partial-wave quantum numbers $L$ up to 4. For collisions involving excited-state atoms, the scattering length is complex, $a = \alpha - i\beta$, and the 2-body loss-rate coefficient at limitingly low energy is $k_2 = (4g\pi\hbar/\mu)\beta$, where $\mu$ is the reduced mass and $g$ is 1 (2) for distinguishable (indistinguishable) particles. Figure 1 shows the real parts of the s-wave scattering lengths $a_{m_{f1},m_{f2}}$ for the three possible combinations of atoms initially in $m_f = 3$ and $m_f = 2$, and the corresponding rate coefficients $k_2$ for two-body inelastic loss (which cannot occur for two atoms with $m_f = 3$). As is well known, the Cs scattering lengths are strikingly field-dependent, featuring overlapping broad and narrow Feshbach resonances. The state (3, 3) features a comparatively gentle zero crossing near 17 G, and BEC in this state has been achieved in a narrow window around 21 G[3]. The state (3, 2) exhibits a broad s-wave resonance centered at 102 G and two gentle zero crossings near 35 G and 148 G. Its magnetic-field dependence is scarred by a multitude of narrow d-wave ($L = 2$) and g-wave ($L = 4$) Feshbach resonances. In fact, the zero crossing near 148 G is split in two by a narrow d-wave resonance. The two-body loss-rate coefficient $k_2$ is strictly zero for the state (3, 3). This is not so for the state (3, 2), but even here spin-exchange

collisions, which conserve $m_{f1} + m_{f2}$, are energetically forbidden at low energies. The two-body loss rates in Fig. 1 are due entirely to spin-relaxation collisions, driven by the magnetic dipole-dipole interaction and second-order spin-orbit coupling[28,30], and there are windows near 40 G and 160 G where the values for (3,2)+(3,2) are small or even negligible. It is these windows on which we will concentrate in this work.

## Results

### BEC of Cs in the state (3,2)

The procedure used to achieve a condensate in the state (3, 2) makes use of some of the tricks that have previously been used to create a BEC in the ground state (3, 3)[4]. We start by loading about $2.5 \times 10^8$ atoms into a six-beam magneto-optical trap (MOT) within 4 s from a Zeeman-slowed atomic beam. Subsequent Raman-sideband cooling in the presence of a near-detuned optical lattice for a duration of 6.9 ms brings the atoms to temperatures below 1 $\mu$K and spin-polarizes them into the state (3, 3). The sample, now with about $5 \times 10^7$ atoms, is loaded into a large-volume "reservoir" dipole trap by gradually switching off the lattice light as a levitating magnetic quadrupole field of 31.1 G/cm is turned on while the magnetic field $B$ is ramped to $B = 160.3$ G, at which $a_{3,3} = 1500$ $a_0$. The stability of the magnetic field is approximately 30 mG. The trap is generated by two horizontally propagating laser beams at 1064.5-nm, intersecting at nearly a right angle. The sample is held for 500 ms to allow plain evaporation at a trap depth of about 2.6(2) $\mu$K $\times k_B$. We now have about $6.5 \times 10^6$ atoms. To transfer the atoms into the state (3, 2) we use a radio-frequency sweep across a range from 54.6 to 54.2 MHz with a duration of 1.45 ms. The levitating field is increased during the sweep to 46.65 G/cm to levitate the atoms in the state (3, 2). The state-transfer efficiency that we can obtain is about 75%. We attribute this to the motional excitation of the atoms as they see changing forces, leading to some heating and hence loss in the finite-depth optical trap. We now have about $4.9 \times 10^6$ atoms at a temperature of around 1 $\mu$K with a peak density of $8.2(1) \times 10^{10}$ atoms/cm$^3$. We estimate the peak elastic collision rate to be 2.8 /s, with $a_{2,2} = 274$ $a_0$ at $B = 160.3$ G. The geometrically averaged trap frequency is $\bar{\nu} = 8.1(5)$ Hz.

Next, as the final step towards BEC in (3, 2), the sample is loaded into a tighter "dimple" trap generated by two orthogonally intersecting 1064.5-nm laser beams with estimated $1/e^2$-waist sizes of 40 and 150 $\mu$m, respectively, one propagating horizontally along the same axis as one of the reservoir beams, and the other propagating vertically. The loading process is completed after 1.5 s, and we then carry out forced evaporative cooling for 6.0 s by lowering the power of both dimple-trap beams in an approximately exponential manner. At the beginning of the evaporation process, the scattering length is tuned to $a_{2,2} = 255$ $a_0$, by ramping the offset field to $B = 159.1$ G. At this value, a minimum for the loss is found, similar to the Efimov minimum for the state (3, 3)[5]; it is crucial to utilize this minimum for optimal performance of the cooling process. The phase transition to a BEC occurs at a critical temperature of about 82(1) nK with $9.5 \times 10^4$ atoms after approximately 3 s of forced evaporation. Absorption images and horizontally integrated density profiles across the transition are shown in Fig. 2a. The evolution via a characteristic bimodal distribution can clearly be seen. The images are taken after 46 ms of time-of-flight (TOF) with nulled interactions upon release[3] by means of the zero crossing in $a_{2,2}$ near 148 G, due to a Feshbach resonance near 102 G. At the end of the evaporation ramp, we obtain an essentially pure BEC with approximately $3.0 \times 10^4$ atoms. The condensate fraction is above 90%. At this point, the dimple trap has trapping frequencies $(\nu_x, \nu_y, \nu_z) = (4.2(3), 6.5(2), 4.9(1))$ Hz, with a trap depth of $V = 8.0(2)$ nK $\times k_B$. The peak density in the Thomas-Fermi (TF) regime is estimated to be $3.6(1) \times 10^{12}$ atoms/cm$^3$, and the TF radii are calculated to be $(R_x, R_y, R_z) = (18.5(2), 15.5(1), 17.4(1))$ $\mu$m. The BEC is comparatively stable, with an atom-number $1/e$-lifetime of around 30 s, most likely

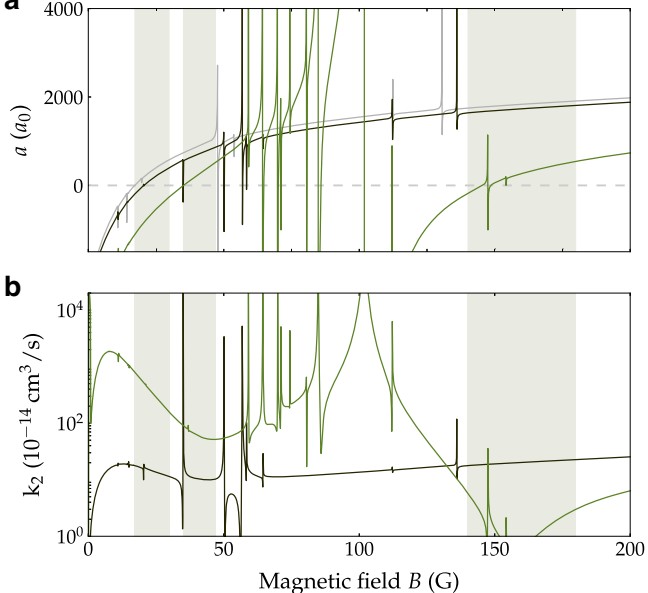

**Fig. 1 | Two-body scattering properties of the Cs states of interest from coupled-channel calculations. a** The real parts of the scattering lengths $a_{3,3}$ (gray), $a_{3,2}$ (black), and $a_{2,2}$ (green) and **b** the two-body loss-rate coefficients $k_2$ for collisions of (3, 3) with (3, 2) (black) and (3, 2) with (3, 2) (green) as a function of the magnetic field $B$. The regions of interest for this work are indicated by the gray shadings.

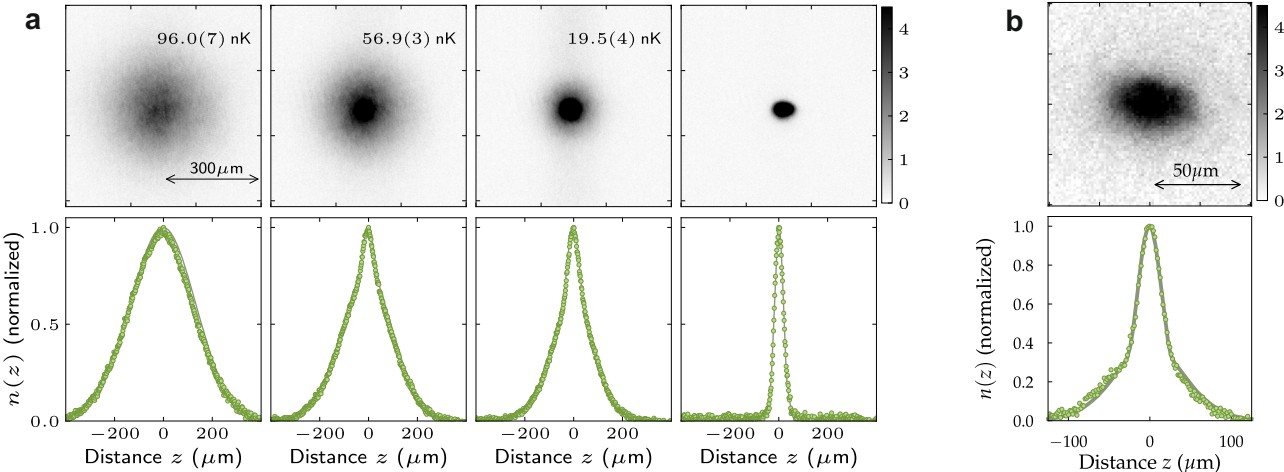

**Fig. 2 | Formation of a BEC in (3, 2) at 160 G and partial BEC at 40 G. a** Absorption images at 160 G (top row), and the resulting horizontally integrated density profiles (bottom row), for different times during the evaporation ramp. We indicate the corresponding temperatures in the absorption images. The images are taken after release from the trap and subsequent 46 ms of TOF with nulled interactions. Each image is an average of five realizations. Bimodal fits to the normalized density profiles give the temperatures as indicated. The final BEC contains about $3.0 \times 10^4$ atoms. **b** Absorption image of the partial BEC at 40 G (top) and normalized vertical density profile (bottom) fitted with a bimodal distribution, after a TOF of 96 ms. The sample has an atom number of $N = 7 \times 10^3$ with a BEC fraction of approximately 30%. These measurements are the averages of four repetitions. We attribute the slight asymmetry that can be observed in the integrated $z$-profile to the spilling of atoms into the vertically propagating dipole-trap beam.

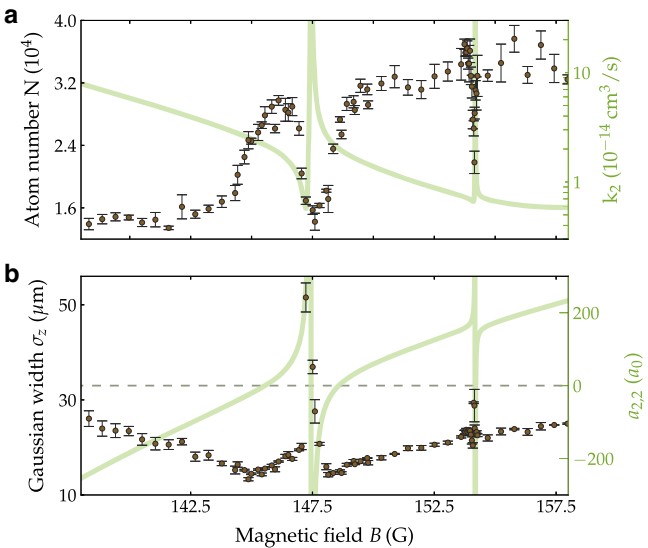

**Fig. 3 | Exploring the vicinity of the zero crossing near 148 G with the BEC in (3, 2). a** Number of atoms $N$ (circles) and **b** the Gaussian cloud width $\sigma_z$ (circles) of the BEC after TOF for different values of the magnetic field $B$. The experimental results in **a** and **b** are overlayed by the calculated two-body loss-rate coefficient $k_2$ (on a log scale) and scattering length $a_{2,2}$, respectively (green lines). Each data point is the average of at least three measurements, and the error bars give the standard error.

limited by slightly imperfect vacuum conditions and residual trap-light scattering. Overall, the BEC in (3, 2) performs nearly as well as the BEC in (3, 3). The cycle time for creating the BEC is 20 s.

We now turn to the window near 40 G. We have not been able to create a BEC in this window by means of the sequence outlined above, with the difference that the magnetic offset field $B$ is ramped to this window at the beginning of the reservoir-trap stage; this is due to the losses discussed below. However, we are partially successful by implementing the lattice trick: The BEC in state (3, 2), created by the sequence above, is adiabatically loaded into a 3D optical lattice at 1064.5 nm with a depth of 25 $E_r$, where $E_r$ is the photon-recoil energy. The lattice is the same as in some of our previous works; see Ref. 11,12. By adjusting the

confinement via the dimple-trap beams we create a Mott insulator with predominant single-site occupancy. The lattice shields the atoms from collisions as the bias field is ramped from 160 G to a value near 40 G in 0.5 ms. Atom loss and sample heating are found to be negligible during the ramp, but they immediately set in when the lattice is unloaded and the atoms are released into the 3D dimple trap. Nevertheless, further evaporative cooling yields BECs of around $7.0 \times 10^3$ atoms with condensate fractions of up to 30%, as seen in Fig. 2b. The lattice trick can be similarly implemented to transfer a BEC in the state (3, 3) to the state (3, 2) at 40 G directly, but without improving the BEC fraction.

### Exploring the zero crossing around 148 G

With a BEC in (3, 2) at hand we now explore the magnetic tunability of the state (3, 2). We focus on the zero crossing of $a_{2,2}$ around $B = 148$ G, which is caused by the broad resonance at 102 G. This zero crossing is of particular interest, not just because of its shallow nature, but also because of the existence of a narrow resonance in its close vicinity. This resonance is decayed, so it does not produce a pole in the scattering length[31]; instead, it produces a sharp oscillation in the scattering length and an asymmetric peak in the two-body loss rate centered at $B_{res} = 147.44$ G, as shown in Fig. 3. It thus gives rise to two zero crossings. We have characterized this resonance from coupled-channel calculations using the methods of Ref. 32, and obtain amplitude $a_{res} = 5.8 \times 10^6 a_0$ and strength $a_{bg}\Delta = 54$ G $a_0$; see Supplementary Note 1 for further details. This setting provides an ideal playground to quench a non-interacting system into a highly interacting one in a fast but controlled manner.

For the measurements we start with pure BEC in (3, 2) at $B = 160.3$ G with a calculated scattering length $a_{2,2} = 274 \, a_0$. We switch off the optical trap to initiate TOF while keeping the levitating field on. Within 0.2 ms $B$ is ramped to the target value, where we allow the sample to expand for another 110 ms. All magnetic fields are switched off and 6 ms later an absorption image is taken, from which we determine the number of remaining atoms $N$ and the cloud width $\sigma_z$ of the atomic sample along the vertical direction $z$. The results are shown in Fig. 3. Both data sets show sharp features due to Feshbach resonances on top of a varying background. Significant atom loss happens for values of $B$ below about 146 G and in the vicinity of the resonances; the resonant loss peaks are centered at 147.62(3) G and 154.13(1) G.

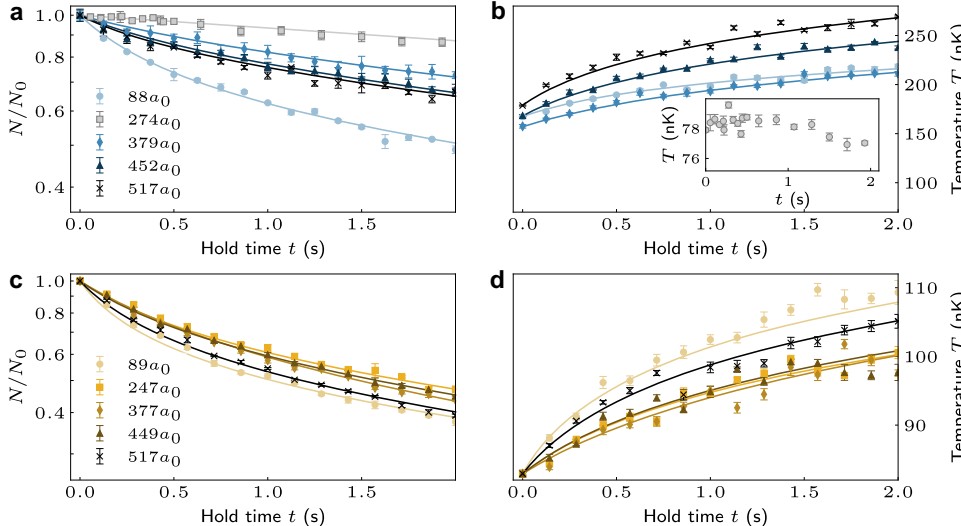

**Fig. 4 | Atom-loss and temperature measurements for non-condensed samples in state (3, 2). a, c** Normalized number of remaining atoms $N$ and **b, d** temperature $T$ for varying hold time $t$ in the region around 160 G (40 G). For **a** and **b**, the scattering length $a_{2,2}$ is set to 88 $a_0$ (circles), 274 $a_0$ (squares), 379 $a_0$ (diamonds), 452 $a_0$ (triangles), and 517 $a_0$ (crosses) for $B = 151.1$ G, 160.3 G, 167.0 G, 172.4 G, and 177.7 G, respectively. The inset in b) shows the measured temperature for the sweet spot with $a_{2,2} = 274$ $a_0$. For **c** and **d** $a_{2,2}$ is set to 89 $a_0$ (circles), 247 $a_0$ (squares), 377 $a_0$ (diamonds), 449 $a_0$ (triangles), and 517 $a_0$ (crosses) for $B = 37.1$ G, 40.7 G, 43.7 G, 45.4 G and 47.0 G, respectively. Each data point is equal to the mean of three to five repeats, and the error bars reflect the standard error. The solid lines fit the data as discussed in the main text. The loss-rate coefficients obtained from the fits are given in Table 1.

The d-wave resonance causes two zero crossings in the calculated scattering length, at 145.5 G and 148.6 G. The BEC shows the highest stability when the value of $a_{2,2}$ is non-zero and positive, as expected from mean-field theory. However, for negative values of $a_{2,2}$ mean-field theory predicts a collapsing BEC. In fact, below the zero crossing at 145.5 G, as $B$ is lowered and $a_{2,2}$ becomes more negative, we observe more loss and increased sample widths. The presence of the two zero crossings is reflected in two minima in the cloud widths at 144.9(1) G and 148.1(1) G. These are at slightly lower fields than the zero crossings, i.e., at slightly negative scattering lengths. Evidently, slightly attractive interactions lead to reduced widths in expansion. This we attribute to the fact that the atom clouds contract for small attractive interactions but do not fully collapse. The increased loss close to the resonances is caused by a combination of resonantly enhanced two-body and resonantly enhanced three-body losses. To sort out which of these dominates near the resonances would need a separate investigation.

Fitting the two resonant loss features with simple Gaussians gives resonance positions at 147.62(3) G and 154.13(1) G. These values are in good agreement with the predictions. Further, more narrow resonant features, which do not appear in Fig. 3 because of the finite resolution of the scan over $B$, are discussed in Supplementary Note 2.

## Measurement of loss-coefficients

The usability of a BEC in (3, 2) in future experiments is, in part, determined by the extent to which the BEC is compromised by losses. We therefore investigate the loss dynamics of ultracold non-condensed samples of atoms in (3, 2) for different values of $a_{2,2}$ in the regions around both 160 G and 40 G. Atoms in the state (3, 2) are exposed to both two-body and three-body decay channels. Two local minima in $k_2$ are predicted around 40 G and 160 G, as seen in Fig. 1. The value for the minimum at 160 G is calculated to be nearly two orders of magnitude lower than that for the minimum near 40 G. A distinct difference in the inelastic scattering behavior around 40 G and 160 G is clearly reflected by the fact that a pure BEC can be achieved at 160 G, but not near 40 G. For three-body recombination there are no precise predictions for these regions. Our lifetime measurements as discussed in this section provide an experimental estimate for both the two- and three-body coefficients in these regions.

We first focus on the region around 160 G. On the way to condensation, we stop the evaporation process and create non-condensed samples in the state (3, 2) in the crossed dimple trap at a depth of approximately 3.1 $\mu$K. At this stage, the samples have a temperature of 500 nK. For loss measurements away from the "sweet spot" with $B = 159.1$ G and $a_{2,2} = 274$ $a_0$ discussed above, we further use "tilt" cooling[33] to decrease the temperature of the sample to around 150 nK without significantly affecting the trap curvature. Tilting the trap back gives such a deep trap that evaporative losses are minimized during the loss measurements, increasing our sensitivity for a potential measurement of $k_2$. With trapping frequencies of $(\nu_x, \nu_y, \nu_z) = (111.0(19), 118.1(38), 40.3(6))$ Hz, we get an initial peak density of the cloud in the range of $1.2 - 1.6 \times 10^{13}$ cm$^{-3}$ for typically $5 \times 10^4$ atoms. We ensure that the sample remains above the BEC phase-transition temperature in order to simplify the modeling of our measurements. In the experiment, we hold the sample for a variable hold time $t$ and then determine atom number and temperature. The results are presented in Fig. 4 a) and b) for hold times up to 2 s and for 4 different values of $B$ from 151.1 to 177.7 G. Particle loss and sample heating are evident. For comparison, we add a data set that is taken at the sweet spot ($B = 159.1$ G and $a_{2,2} = 274$ $a_0$) without using the tilt method but just by stopping the evaporation sequence shortly before condensation. Here, some loss can be seen, but no heating. The loss is most likely evaporative loss, and possible heating is balanced by plain evaporative cooling, given the rather shallow trap. For the other data sets, there is an obvious trend that larger values of $a_{2,2}$ result in faster loss and a more rapid temperature increase. However, we find fast loss also for lower values of $a_{2,2}$ away from the sweet spot, as can be seen from the data set with 88 $a_0$.

To model the loss, we assume that the samples remain in thermal equilibrium throughout the whole process. The number $N$ of remaining atoms evolves according to the rate of change of the density, $\dot{n}(\mathbf{r}, t) = -\sum_i k_i n(\mathbf{r}, t)^i$, where $i = 1, 2$, and 3 denote the one-, two-, and three-body loss processes, respectively, with $k_i$ the coefficient of the collision rate for $i$-body loss. The rate of change for the atom number is obtained by integrating $\dot{n}(\mathbf{r}, t)$, $\dot{N}(t) = \int \dot{n}(\mathbf{r}, t) d^3 \mathbf{r}$. As is well known[27], the atom loss from the trap induces heating via two dominant processes. These are known as anti-evaporation and recombination heating. Following Ref. 27 we incorporate the latter by an additional

**Table 1 | Experimentally determined and theoretically predicted values of loss-rate coefficients in the windows around 40 G and 160 G**

| B (G) | $a_{2,2}(a_0)$ | $k_1$ (s⁻¹) | Experiment $k_2$ (10⁻¹⁴ cm³ s⁻¹) | $k_3$ (10⁻²⁶ cm⁶ s⁻¹) | Theory $k_2$ (10⁻¹⁴ cm³ s⁻¹) |
|---|---|---|---|---|---|
| 37.1 | 89 | - | 18.2(49) | 323.2(108) | 73.2 |
| 40.7 | 247 | - | 35.5(25) | 179.9(47) | 59.2 |
| 43.7 | 377 | - | 88.7(30) | 52.9(55) | 47.0 |
| 45.4 | 449 | - | 49.5(27) | 135.2(53) | 51.8 |
| 47.0 | 517 | - | 39.4(26) | 156.5(44) | 51.6 |
| 151.1 | 88 | 0.040(11) | 2.1(12) | 4.56(22) | 1.2 |
| 167.0 | 379 | 0.057(12) | 1.2 (fixed) | 0.50(10) | 1.2 |
| 172.4 | 452 | 0.062(1) | 1.8 (fixed) | 1.11(2) | 1.8 |
| 177.7 | 517 | 0.054(4) | 2.3 (fixed) | 1.86(5) | 2.3 |

The errors in the experimental values are determined from the fit.

temperature parameter $T_h$ in the three-body heating term of the rate equation. To obtain the loss-rate coefficients, the data points are then fitted by the numerical solution of the resulting coupled rate equations (Eq. (1) and Eq. (2) of the Methods). The fits are shown along with the experimental data in Fig. 4, and the loss-rate coefficients obtained are summarized in Table 1. The model fits the data reasonably well, with the two-body loss rate being negligible and the loss thus dominated by three-body recombination. The one-body loss rate coefficient $k_1$ is ~ 0.05 s⁻¹ for all the data sets. In cases where the fitted values of $k_2$ are close to zero, we fix them to the corresponding theoretical value to avoid overfitting. Note that the value for $k_1$ is significantly larger than we obtain for a pure BEC, where we use a much lower laser power for the dipole trap. We attribute the need for a higher value of $k_1$ here to increased losses due to inelastic light scattering. Otherwise, the values obtained for $k_3$ for a given value of $a_{2,2}$ are similar to the ones from previous work using the sublevel (3, 3)[5,27].

We now turn to the region around 40 G. The measurements here have a surprise for us. This region is again reached using the lattice trick. The trap depth is approximately 800 nK, with an initial temperature of the samples between 80 and 90 nK and with peak densities of around $1.5 × 10^{12}$ cm⁻³. Note that this time the density is a factor of 10 lower than in the previous measurements. Figures 4c and d show the resulting loss and heating, together with fits similar to those above. First of all, no sweet spot can be identified. For the plot, the values for $B$ were chosen such that the values for $a_{2,2}$ are the same as for the measurements in the region around 160 G. Evidently, accounting for the lower density, the loss and heating observed is significantly greater than in the region around 160 G. The decay curves lie closer together, i.e., the loss does not depend so much on the value of $a_{2,2}$. All this is reflected by the fit results. When testing the fits, we find that both two- and three-body loss are significant. However, leaving all parameters, i.e., $k_1$, $k_2$, $k_3$ and $T_h$, as free parameters makes our model prone to overfitting. We fix $k_1 = 0.068$ /s, and we omit the contribution of $T_h$ from the fits as it appears to be negligible. Then, letting $k_2$ and $k_3$ vary freely, we find values for $k_2$ of about $5 × 10^{-13}$ cm³ s⁻¹ and for $k_3$ between about $0.5 × 10^{-24}$ cm⁶ s⁻¹ and $3.2 × 10^{-24}$ cm⁶ s⁻¹. The specific values are added to Table 1. The values for $k_2$ agree reasonably well with the theoretical values, but the values for $k_3$ are nearly two orders of magnitude larger than we obtained from the measurements in the region around 160 G or from measurements involving the state (3, 3)[5,27], at given values for the relevant scattering length away from the sweet spots (which are at 160 G for (3, 2) and at 21 G for (3, 3)).

Such high values for $k_3$ are a surprise, and we can only speculate about the origin of such high three-body loss. A simple calculation (given in Supplementary Note 3) shows that there is a possible three-body recombination process that flips the spin of one atom to (3, 1) while forming a molecule in the least-bound state of the channel (3, 3) + (3, 2). This process becomes energetically resonant near $B = 45$ G due to the second-order Zeeman effect. Three-body recombination processes assisted by spin exchange are believed to be important for ⁷Li[34,35] and ³⁹K[36,37], but not for ⁸⁷Rb[38,39] and ⁸⁵Rb[40]. We note that this process does not seem to play a role in the window near 160 G. To discuss the likelihood of this explanation further, we consider the energy mismatch between three free particles and the products of the three-body recombination. In ref. 34, the presence of an additional spin channel, with an energy mismatch of approximately $0.25 E_{vdW}$ was proposed as the cause of a discrepancy of about two orders of magnitudes in the measured $k_3$. Here $E_{vdW}$ is the typical energy scale for the van der Waals interaction between the two atoms[2]. The energy mismatch in our case is significantly smaller, in the range of 0 to $0.05 E_{vdW}$.

## Discussion

In summary, for the first time, a BEC of Cs has been obtained in a spin state other than the absolute ground state (3, 3). We have identified two windows where this is possible: In a region around 160 G, two-body losses are negligible, and three-body losses are sufficiently suppressed. Here, a pure BEC with $3.0 × 10^4$ atoms in (3, 2) can be formed.

The situation is different in a region around 40 G. We have not been able to achieve BEC by direct evaporation in this region. However, creating a BEC first near 160 G and then transferring it in a lattice to 40 G allows us to produce BECs at that field value. Losses are strong, reducing the purity of the BEC and limiting its lifetime to around 0.5 s. Interestingly, three-body rather than two-body losses are the limiting factor. We have not been able to find a sweet spot where three-body recombination is sufficiently suppressed. In fact, the three-body loss-rate coefficient is close to two orders of magnitude larger than we had expected; the high value may be the result of spin-flip-aided three-body recombination. This process merits further investigation. One signature would be the direct detection of atoms in (3,1) produced in the recombination process. We note that the losses in this region set in only when the atoms are released from the lattice into the 3D trap. The losses are nearly fully suppressed when the atoms are kept in, e. g., 1D tubes, as has been done previously in various experiments in our group[10,41].

More than 20 years after the first attainment of a Cs BEC in (3, 3)[3], BEC in (3, 2) is not merely an academic achievement. While Cs spinor BECs remain out of reach, BEC in (3, 2) opens up new possibilities for impurity and polaron physics. Specifically, for experiments in the context of strong bulk-bulk and impurity-bulk correlations in 1D[12], the strengths of the bulk-bulk and the impurity-bulk interactions can be interchanged, potentially allowing the impurity to serve as a matter-wave probe of the pinning transition[11] through the phase transition point. Precision tests of the underlying quantum field theory, the sine-Gordon model[42], are thus possible. Further uses of (3, 3) as a strongly interacting probe (instead of (3, 2)) will enhance experiments on topological phase transitions[43]. In addition, being able to condense Cs in (3, 2) opens new possibilities in quantum-gas mixture setups, e.g., for the production of ultracold and possibly quantum-degenerate samples of heteronuclear molecules such as KCs[44] and RbCs[45,46].

## Methods
### Fitting of loss measurements
For our fitting procedure, we use a coupled fit of the atom number and the temperature evolution[27,46]

$$\dot{N}(t) = -k_1 N(t) - k_2\beta \frac{N(t)^2}{2^{3/2}T(t)^{(3/2)}} - k_3\beta^2 \frac{N(t)^3}{3^{3/2}T(t)^3}, \quad (1)$$

$$\dot{T}(t) = k_2\beta \frac{N(t)}{2^{7/2}T(t)^{(1/2)}} + k_3\beta^2 \frac{N(t)^2(T(t)+T_h)}{3^{5/2}T(t)^3}, \quad (2)$$

where $\beta = (m\bar{\omega}^2/2\pi k_B)^{3/2}$ with the mass $m$ of the Cs atom, and $\bar{\omega} = 2\pi \times \bar{\nu}$ is the geometrically averaged trap frequency. Applying the standard approach of the least-square fit method we minimize the function

$$\chi^2 = \sum_i \left(\frac{r_N(i)}{\sigma_N(i)}\right)^2 + \sum_i \left(\frac{r_T(i)}{\sigma_T(i)}\right)^2. \qquad (3)$$

Here $r_N = (N_{exp} - N_{mod})$ $(r_T = (T_{exp} - T_{mod}))$ is the residual of the number of atoms (temperature), $N_{exp}$ $(T_{exp})$ is the measured atom number (temperature), and $N_{mod}$ $(T_{mod})$ is the corresponding value obtained from Eq. (1) (Eq. (2)). The weighted error of the atom loss (temperature) measurements is $\sigma_N(i)$ $(\sigma_T(i))$. For the region near 160 G we use $k_1$, $k_3$ and $T_h$ in Eqs. (1) and (2) as free parameters. Additionally using $k_2$ as a free parameter yields non-zero values of $k_2$ only for the data set at 151.1 G. We hence leave $k_2$ as a free parameter for this particular data set and set it to the corresponding theoretical values for the others. In the region near 40 G, we leave $k_3$ and $k_2$ as free parameters and restrict $k_1 = 0.068$ /s and $T_h = 0$ $\mu$K to avoid overfitting. Note that for all of these fits, the correlation coefficients between the loss coefficients are above 0.9. The fitting parameters listed in Table 1 include the standard error, which is derived from the diagonal elements of the variance-covariance matrix.

## Data availability

Data supporting this study are openly available from Zenodo at[47].

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

## Acknowledgements
We thank R. Grimm for discussions and for pointing out to us the possible loss mechanism in the region near 40 G. The Innsbruck team acknowledges funding by a Wittgenstein prize grant under the Austrian Science Fund's (FWF) project number Z336-N36, by the European Research Council (ERC) under project number 789017, and by an FFG infrastructure grant with project number FO999896041. MH thanks the doctoral school ALM for hospitality, with funding from the FWF under the project number W1259-N27. The theoretical work was supported by the UK Engineering and Physical Sciences Research Council (EPSRC) Grant Nos. EP/P01058X/1, EP/V011677/1 and EP/W00299X/1.

## Author contributions
M.H. and S.D. performed the experiments. M.H., S.D., and A.D. analyzed the experimental data. M.D.F. carried out the scattering calculations. H.-C.N., M.L., and J.M.H. supervised the project. M.H., S.D., A.D., M.D.F., Y.G., M.L., J.M.H., and H.-C.N. contributed to the writing of the manuscript.

## Competing interests
The authors declare no competing interests.
