## [Peer Review File · Nature Communications]

Bose-Einstein condensation of non-ground-state caesium atomsReviewer #1 (Remarks to the Author):

The manuscript, "Bose-Einstein condensation of non-ground-state cesium atoms" by Horvath et al. reported the realization of the Bose-Einstein condensation (BEC) of cesium atoms in the excited state. The authors showcased and analyzed the stability of the sample, and measured the loss rate of the BEC in a controlled manner. The result is clearly presented in the manuscript and the relevant experimental details are well discussed.

A state of BEC is a platform where various quantum phenomena, such as superfluidity, can be hosted (when interactions exist, which is the case for cold atoms). Over the past years, several groups have been able to manage ultracold atoms in an excited level, including the higher orbitals in optical lattices, meta-stable states, and hyperfine levels, which enables us to access unconventional superfluid (e.g chiral superfluid), two-orbital physics or spinor physics. Along this path, the reported work is intriguing, offering a new opportunity to explore novel quantum phases. The property of the Cs BEC in an excited state is seemingly stable with a long enough lifetime, and one can imagine that various many-body physics can be tested with this sample. Therefore, in principle, the manuscript can be published in some forums, but several major/minor points need to be adequately addressed before I recommend the publication in Nature Communications.

(1) In my opinion, it is recommended to broaden the scope of the introduction. As I described above, the reported work is of broad importance, but the current version does not highlight this aspect.

(2) The collisional property has been comprehensively investigated in 3D. What about in low dimensions? Can authors provide more discussion on this?

(3) As far as I know, two-body spin exchange collisions are very low in the Cs system due to the large quadratic Zeeman shift. This means that the two-body loss process is dominantly coming from the spin relaxation. In the main text, it is a bit confusing about this point. The authors are recommended to make this point clear.

(4) In the introduction, authors mentioned Sr/Yb spinor BECs. It is confusing that Sr and Yb BECs are spinless. Could authors clarify this point?

Reviewer #2 (Remarks to the Author):

The authors experimentally created a Cs BEC in the Zeeman-excited state (3, 2), and observed the different collision processes in two windows around 40G and 160 G. Specifically the large three-body loss in the window near 40 G is surprising. The manuscript is carefully written and easy to be flowed. However, there are several points that should be clarified before I reconsider this paper for publication in Nature Communications. They should address the following comments and questions.

(1). The main reason that induced the different collisions in the two windows is the three-body process. The authors measured the three-body loss rate in the two windows, listed in Table I. However the speculation about the three-body loss rate is most likely due to the opening up of a new decay channel, and the explanation on page 6 is too easy. I hope to read a convincing explanation or some effort in this direction.

(2). In Table I, the loss rates have been listed. The rate in 151.1 G makes me confused because of the bigger two-body loss rate, compared to the theory value and the other values in the window around 160 G. It would be better to show some explanations.

(3). The authors cited the works of Rb, Na, Sr, and Yb in the introduction, which has been prepared in BEC in the Zeeman-excited state. I don't know why they lost 39K with rich Feshbach structures, have been created in each one of the three hyperfine states of $F=1$ manifold. The authors should cite the papers of 39K.

Response to the referees' comments on the manuscript: Bose-Einstein condensation of non-ground-state caesium atoms

We thank the two reviewers for their constructive comments, which have allowed us to improve the manuscript. In the following, we itemize detailed responses to the reviewers' comments (all reviewers' comments are in blue).

Response to reviewer 1:

Reviewer 1's comment: The manuscript, "Bose-Einstein condensation of non-ground-state cesium atoms" by Horvath et al. reported the realization of the Bose-Einstein condensation (BEC) of cesium atoms in the excited state. The authors showcased and analyzed the stability of the sample, and measured the loss rate of the BEC in a controlled manner. The result is clearly presented in the manuscript and the relevant experimental details are well discussed.

A state of BEC is a platform where various quantum phenomena, such as superfluidity, can be hosted (when interactions exist, which is the case for cold atoms). Over the past years, several groups have been able to manage ultracold atoms in an excited level, including the higher orbitals in optical lattices, meta-stable states, and hyperfine levels, which enables us to access unconventional superfluid (e.g chiral superfluid), two-orbital physics or spinor physics. Along this path, the reported work is intriguing, offering a new opportunity to explore novel quantum phases. The property of the Cs BEC in an excited state is seemingly stable with a long enough lifetime, and one can imagine that various many-body physics can be tested with this sample. Therefore, in principle, the manuscript can be published in some forums, but several major/minor points need to be adequately addressed before I recommend the publication in Nature Communications.

1. In my opinion, it is recommended to broaden the scope of the introduction. As I described above, the reported work is of broad importance, but the current version does not highlight this aspect.

Our response: We have accordingly adapted the introduction with a new paragraph, highlighting the general interest of such systems and added a citation to the work on chiral superfluidity.

2. The collisional property has been comprehensively investigated in 3D. What about in low dimensions? Can authors provide more discussion on this?

Fig. 1. **Atom loss in one-dimension for different interactions.** Normalized atom number as a function of time where $a_{2,2}$ is set to $166 a_0$ (circle), $279 a_0$ (square), $479 a_0$ (diamond), and $987 a_0$ (triangle) for $B = 38.8$ G, 41.4 G, 46.6 G, 56.9 G and 47.0 G, respectively. Each data point represents the average of two repetitions. The solid line are exponential fits of the data.

Our response: In general, this is a question of great interest to us. We have investigated the collisional properties of a (3,2) gas in lower dimension. As mentioned in the concluding remarks of the main text, typically losses get suppressed as dimensions are reduced.

For example, in the 40 G region, high three-body losses have prevented us from obtaining a pure condensate. However, the situation is different in 1D. Here, we can create a relatively stable 1D Bose gas in this region by employing the lattice-trick as mentioned in the main text. In Fig. 1 we show the atom number as a function of time for different interactions in a 1D system around 40 G. We fit the data with simple exponential functions with an offset, which gives us lifetimes between 200 – 1000 ms. The stability of the system increases with increasing interactions. Similar behavior has been previously observed in references [1, 2]. There it is shown that as the interactions are increased, due to emergent Pauli blocking, both second and third order correlation functions at zero distance $g_2(0)$ and $g_3(0)$ approach zero leading to this effect. Around 160 G the gas is collisionally stable in every dimension with lifetimes more than 20 s.

Due to the similarity of behavior already investigated in detail in previous studies we have refrained from adding this discussion to the main text.

3. As far as I know, two-body spin exchange collisions are very low in the Cs system due to the large quadratic Zeeman shift. This means that the two-body loss process is dominantly coming from the spin relaxation. In the main text, it is a bit confusing about this point.

The authors are recommended to make this point clear.

Our response: The referee is correct and we have clarified this point in the description of the scattering calculations. The relevant part now reads: “The two-body loss-rate coefficient k_2 is strictly zero for the state (3,3). This is not so for the state (3,2), but even here spin-exchange collisions, which conserve $m_{f1} + m_{f2}$, are energetically forbidden at low energies. The two-body loss rates in Fig. 1 are due entirely to spin-relaxation collisions, driven by the magnetic dipole-dipole interaction and second-order spin-orbit coupling [3, 4], and there are windows near 40 G and 160 G where the values for (3,2)+(3,2) are small or even negligible.”

4. In the introduction, authors mentioned Sr/Yb spinor BECs. It is confusing that Sr and Yb BECs are spinless. Could authors clarify this point?

Our response: There are attempts in the literature to use the metastable triplet states in (bosonic) Sr and Yb for quantum gas physics. It turns out the lifetimes are unfortunately too short for getting interesting results. We have appropriately removed the mention of Sr and Yb gases.

Response to reviewer 2:

Reviewer 2’s comment: The authors experimentally created a Cs BEC in the Zeeman-excited state (3, 2), and observed the different collision processes in two windows around 40G and 160 G. Specifically the large three-body loss in the window near 40 G is surprising. The manuscript is carefully written and easy to be flowed. However, there are several points that should be clarified before I reconsider this paper for publication in Nature Communications. They should address the following comments and questions.

1. The main reason that induced the different collisions in the two windows is the three-body process. The authors measured the three-body loss rate in the two windows, listed in Table I. However the speculation about the three-body loss rate is most likely due to the opening up of a new decay channel, and the explanation on page 6 is too easy. I hope to read a convincing explanation or some effort in this direction.

Our response: We have added a new section in the supplementary material detailing the resonant behavior of the spin-exchange assisted three-body recombination process. However, this does not give us any information about the loss rate itself. This would require detailed numerical calculations that, to our knowledge, have only recently been carried out for Li.

The study of three-body losses accompanied by spin flips in the participating particles is rather recent and this point certainly deserves further investigation. We believe that such an investigation would go beyond the scope of the present paper. However, we have included a comment at the end of the paragraph, with a quantitative comparison of the energy mismatch expected in our situation and the one present in a recent investigation where the effect of the extra decay channel was found to lead to significant modification of the three-body decay rate.

2. In Table I, the loss rates have been listed. The rate in 151.1 G makes me confused because of the bigger two-body loss rate, compared to the theory value and the other values in the window around 160 G. It would be better to show some explanations.

Our response: The calculation used to determine the error of the loss coefficients was stated incorrectly in the Methods section. The errors reported were determined by changing fitting parameter in question until χ^2 varies by one. However, this method does not take into account the correlations between fitting parameters. Since we have correlations coefficients above 0.90 for most of our fitting parameters, we have updated the error to the standard error from the diagonal of the covariance matrix which does take this correlation into account. This further required an adjustment of the fitting method at 160 G, which we included in both the main text and the Methods section. The experimentally obtained two-body loss coefficient at 151.1 G now falls within error of the theoretically predicted value. We have also added a statement in the Methods section advising the reader of the high correlations between our fitting parameters.

3. The authors cited the works of Rb, Na, Sr, and Yb in the introduction, which has been prepared in BEC in the Zeeman-excited state. I don't know why they lost 39K with rich Feshbach structures, have been created in each one of the three hyperfine states of F=1 manifold. The authors should cite the papers of 39K.

Our response: We have removed the list of atomic species in the introduction, but we have added the citation to the relevant work on quantum droplets of 39K.

[1] E. Haller, M. Rabie, M. J. Mark, J. G. Danzl, R. Hart, K. Lauber, G. Pupillo, and H.-C. Nägerl, Three-Body Correlation Functions and Recombination Rates for Bosons in Three Dimensions and One

- Dimension, Phys. Rev. Lett. **107**, 230404 (2011).
- [2] T. Kinoshita, T. Wenger, and D. S. Weiss, Local pair correlations in one-dimensional bose gases, Phys. Rev. Lett. **95**, 190406 (2005).
- [3] P. J. Leo, C. J. Williams, and P. S. Julienne, Collision Properties of Ultracold ^{133}Cs Atoms, Phys. Rev. Lett. **85**, 2721 (2000).
- [4] M. D. Frye, B. C. Yang, and J. M. Hutson, Ultracold collisions of Cs atoms in excited Zeeman and hyperfine states, Phys. Rev. A **100**, 022702 (2019).

Reviewer #1 (Remarks to the Author):

In the revised version, the authors have successfully addressed all comments/suggestions raised by both referees. In particular, the collisional property of Cs gases has been thoroughly discussed in the context of dimensionality and three-body loss, which is instructive to the reader. I feel the current version is informative with extensive discussion and also readable. The progress demonstrated in the manuscript is not incremental and of timely importance in the community. Therefore, I am happy to recommend the publication in Nature Communication.

Reviewer #2 (Remarks to the Author):

I am satisfied with the answers of the authors. This paper could be published in Nature Communications, as it presents original and useful results.